# Protein Expression of AEBP1, MCM4, and FABP4 Differentiate Osteogenic, Adipogenic, and Mesenchymal Stromal Stem Cells

**DOI:** 10.3390/ijms23052568

**Published:** 2022-02-25

**Authors:** Thorben Sauer, Giulia Facchinetti, Michael Kohl, Justyna M. Kowal, Svitlana Rozanova, Julia Horn, Hagen Schmal, Ivo Kwee, Arndt-Peter Schulz, Sonja Hartwig, Moustapha Kassem, Jens K. Habermann, Timo Gemoll

**Affiliations:** 1Section for Translational Surgical Oncology and Biobanking, Department of Surgery, University Hospital Schleswig-Holstein, University of Luebeck, Campus Luebeck, Ratzeburger Allee 160, 23562 Luebeck, Germany; thorben.sauer@student.uni-luebeck.de (T.S.); facchinetti.giuli@gmail.com (G.F.); michael.kohl@uni-luebeck.de (M.K.); svitlana.rozanova@ruhr-uni-bochum.de (S.R.); julia.horn@uni-luebeck.de (J.H.); jens.habermann@uni-luebeck.de (J.K.H.); 2Department of Endocrinology and Metabolism, University Hospital of Odense, J.B. Winsløws Vej 25, 5230 Odense, Denmark; jkowal@health.sdu.dk (J.M.K.); mkassem@health.sdu.dk (M.K.); 3Department of Orthopedics and Traumatology, Odense University Hospital, Odense, J.B. Winsløws Vej 4, 5000 Odense, Denmark; hagen.schmal@uniklinik-freiburg.de; 4Department of Orthopedics and Trauma Surgery, Faculty of Medicine, Medical-Center—Albert-Ludwigs-University of Freiburg, Hugstetter Straße 55, 79106 Freiburg, Germany; 5BigOmics Analytics SA, 6500 Bellinzona, Switzerland; kwee@bigomics.ch; 6Fraunhofer Research Institution for Individualized and Cell-Based Medical Engineering Luebeck, Moenkhofer Weg 239a, 23562 Luebeck, Germany; schulz@biomechatronics.de; 7BG Klinikum Hamburg, Department Centrum Klinische Forschung, Bergedorfer Str. 10, 21033 Hamburg, Germany; 8German Center for Diabetes Research (DZD), 85764 Muenchen, Germany; sonja.hartwig@ddz.de; 9German Diabetes Center, Heinrich Heine University Duesseldorf, Leibniz Center for Diabetes Research, Institute of Clinical Biochemistry and Pathobiochemistry, 40225 Duesseldorf, Germany; 10Interdisciplinary Center for Biobanking-Luebeck, University of Luebeck, Ratzeburger Allee 160, 23562 Luebeck, Germany

**Keywords:** protein profiling, data-independent acquisition mass spectrometry, SWATH, human stromal/mesenchymal stem cells, differentiation markers, machine learning

## Abstract

Mesenchymal stem cells (MSCs) gain an increasing focus in the field of regenerative medicine due to their differentiation abilities into chondrocytes, adipocytes, and osteoblastic cells. However, it is apparent that the transformation processes are extremely complex and cause cellular heterogeneity. The study aimed to characterize differences between MSCs and cells after adipogenic (AD) or osteoblastic (OB) differentiation at the proteome level. Comparative proteomic profiling was performed using tandem mass spectrometry in data-independent acquisition mode. Proteins were quantified by deep neural networks in library-free mode and correlated to the Molecular Signature Database (MSigDB) hallmark gene set collections for functional annotation. We analyzed 4108 proteins across all samples, which revealed a distinct clustering between MSCs and cell differentiation states. Protein expression profiling identified activation of the *Peroxisome proliferator-activated receptors (PPARs)* signaling pathway after AD. In addition, two distinct protein marker panels could be defined for osteoblastic and adipocytic cell lineages. Hereby, overexpression of AEBP1 and MCM4 for OB as well as of FABP4 for AD was detected as the most promising molecular markers. Combination of deep neural network and machine-learning algorithms with data-independent mass spectrometry distinguish MSCs and cell lineages after adipogenic or osteoblastic differentiation. We identified specific proteins as the molecular basis for bone formation, which could be used for regenerative medicine in the future.

## 1. Introduction

Bone marrow stromal cells (MSCs) were first detected by Friedenstein in murine bone marrow cultures [1] and subsequently gained enormous attention regarding their medical utility including cellular therapy applications. MSCs are non-hematopoietic cells originating from the mesodermal germ layer and are detected in a multitude of tissues including bone marrow. Upon tissue migration, MSCs secrete chemokines, cytokines, and growth factors modulating the immune response, angiogenic as well as anti-apoptotic effects [2,3,4,5]. MSCs have been proven to be effective treatments in many diseases, e.g., cardiovascular diseases [6,7], musculoskeletal diseases [8], neurological diseases [9], immune system defects [10,11], cancer [12] and tissue regeneration in large bone defects [13,14,15,16,17,18]. Their clinical efficacy was recently also tested as a therapeutic approach for patients with a COVID-19 infection [19].

In this study, we focus specifically on the characteristics of MSCs in the context of potential treatment interventions in bone lesions. MSCs can differentiate into osteoblasts (OB), adipocytes (AD), chondrocytes and muscle cells [20]. Although the utilization of MSCs as a cellular therapy approach is clinically investigated, the experimental efficacy of MSCs concerning bone regeneration was reported inconsistently regarding the successful in vivo bone formation [18,21,22,23]. Next to insufficient marker panels for MSC classification, the most likely causes for this observation are cellular and molecular variations of the bone marrow, which does not consist of a homogenous cell type but rather of a population with high cellular heterogeneity containing multipotent stem cells, progenitors and differentiated cells [24,25,26,27]. Furthermore, reports suggest that bone marrow contains clonal MSC subpopulations with associations towards either osteoblast or adipocyte lineage. Hence, MSCs show cellular heterogeneity in the context of in vitro osteoblast differentiation, resulting in heterogenous bone formation capacity in vivo [24,28,29,30,31,32,33,34] and certain bone diseases such as osteoporosis [35,36,37].

Since the nature of the MSC differentiation into osteoblasts and adipocytes remains unclear [23,38,39], the specific objective of this study was to characterize undifferentiated MSCs and cells after osteoblastic and adipocytic differentiation on the proteome level. We utilized quantitative mass spectrometry in data-independent acquisition mode combined with machine-learning algorithms for mass spectrometric and statistical evaluation. Generated quantitative proteome data were used to gain further insight into functional annotation including intracellular signaling pathways and gene ontology terms for potential clinical applications. In the future, identified protein candidates for adipocytes or osteoblasts could be used for regenerative therapeutic approaches healing bone fractures or bone diseases.

## 2. Results

### 2.1. Cultured Cell Populations Fulfilled Phenotypic Criteria of MSC, Adipocytic and Osteoblastic Criteria

The individual MSC isolates (*n* = 5), each derived from one single individual (patients p11, p13, p15, p17, and p18), were characterized in vitro (Appendix A), and have been published elsewhere [32,38]: using a colony-forming unit fibroblast (CFU-F) assay, cultured MSCs formed colonies and expressed alkaline phosphatase (ALP) (median percentage of positive cells ± SD, 37.19% ± 10.91%). The cells were further characterized by their MSC surface marker expression recommended by the International Society for Cellular Therapy (ISCT [39]). Cell material of p11, p15, and p18 were limited, wherefore the determination of all surface markers was only possible for p13 and p17. However, p13 and p17 showed positive expression values for CD44, CD90, CD105 and CD73 in ≥99% (max. SD of 0.16%).

After induction of in vitro osteoblastic differentiation, matrix mineralization was visualized by alizarin red staining and compared to cells cultured in control media (Appendix A). Differentiated cells showed a 3.53-fold higher alizarin red intensity after 14 days of culturing. Additionally, cells showed a 6.35-fold higher ALP activity compared to the control cells. Adipocytic differentiation was assessed by quantifying the lipid droplet area based on Oil Red O staining after 14 days. All cell cultures showed visible lipid droplets after culturing in adipocyte induction media (Appendix A). These results confirmed the successful differentiation of MSCs into adipocytic and osteoblastic cell types, respectively.

### 2.2. Combined Quantitative Mass Spectrometry and Neural-Network-Based Algorithms Revealed Distinct Protein Expression Patterns of MSCs, Adipocytic and Osteoblastic Cells

To compare differences between adipocyte (AD), osteoblast (OB) lineage cells, and undifferentiated MSCs on the proteome level, we performed microflow ESI-MS/MS analysis using the data-independent acquisition mode for exact quantification. The neuronal network-based workflow of DIA-NN identified 4569 unique proteins (FDR ≤ 0.01 on both precursor and protein levels). After filtering and imputation of missing values (cf. Section 4.12), quantitative values of 4108 proteins were subsequently compared between groups (AD, OB, and MSCs) to reveal distinct clustering between samples, differentially expressed proteins, and enriched gene sets (cf. Section 4.13). A table containing all protein quantification values is included in Appendix A.

### 2.3. MSCs, Osteoblasts, and Adipocytes Show Distinct Clustering Behavior and Involvement of PPAR

Phenotypic differences between distinct cell types were visualized using unsupervised principal component analysis (PCA) and hierarchical clustering (so-called ‘heatmap’) applying detected protein expression data. Both, PCA and hierarchical clustering showed a high discriminating potential and clear separation between the three groups (Figure 1A and Appendix A). Analogous to the mass spectrometry-based analysis, two-dimensional gel electrophoresis was performed. Here, 1517 protein spots were detected and revealed a similar clustering behavior in the PCA plot (Figure 1B).

In order to identify overrepresented biological states or processes, a gene set enrichment analysis (GSEA) was performed with mass spectrometry derived expression data of all 4108 quantified proteins against the hallmark gene set collection [40]. Nine gene sets were detected as significantly enriched for the OB/AD, three for the OB/MSC, and 10 for the AD/MSC comparison (FDR < 0.05, |log_2_FC| > 0.2, Table 1 and Appendix A–C). The most striking result to emerge from the data was the association of both AD comparisons with the hallmark gene set collection ‘*adipogenesis*’ which consists of 35 founder gene sets. One of the key signaling pathways for adipogenic differentiation was the ‘*peroxisome proliferator-activated receptor (PPAR) signaling pathway*’ which is exemplarily visualized in the Appendix A including the visualized fold changes of the identified proteins for the AD vs. OB comparison: acetyl-CoA acyltransferase 1 (ACAA1), acyl-CoA dehydrogenase medium chain (ACADM), acyl-CoA oxidase 1 (ACOX1), acyl-CoA synthetase long chain family member 1 (ACSL1), adiponectin, C1Q and collagen domain containing (ADIPOQ), apolipoprotein A1 (APOA1), CD36 molecule (CD36), carnitine palmitoyltransferase 1A (CPT1A), carnitine palmitoyltransferase 2 (CPT2), cytochrome P450 family 27 subfamily A member 1 (CYP27A1), diazepam binding inhibitor, acyl-CoA binding protein (DBI), enoyl-CoA hydratase and 3-hydroxyacyl CoA dehydrogenase (EHHADH), fatty acid binding protein 4 (FABP4), fatty acid desaturase 2 (FADS2), integrin linked kinase (ILK), lipoprotein lipase (LPL), malic enzyme 1 (ME1), phosphoenolpyruvate carboxykinase 1 (PCK1), perilipin 2 (PLIN2), stearoyl-CoA desaturase (SCD), sterol carrier protein 2 (SCP2), solute carrier family 27 member 4 (SLC27A4), sorbin and SH3 domain containing 1 (SORBS1). With the exception of ILK, all proteins show positive fold-change values indicating a prominent pathway activation.

### 2.4. Discovery of Differentially Expressed Proteins

For the detection of differentially expressed proteins, all three two-group comparisons were carried out using an ANOVA with Benjamini–Hochberg correction and Tukey post hoc testing (Appendix A). While the comparison between OB and MSCs identified 50 differentially expressed proteins (post-hoc q-value < 0.05, |log_2_FC| > 2) with 25 proteins being over- and under-expressed in the OB group, respectively (Figure 2a), the comparison OB versus AD revealed 66 proteins with differential abundance (16 over- and 50 under-expressed in the OB group, Figure 2b). The evaluation between AD and MSC revealed 61 proteins with a higher and 25 with lower protein abundance (86 differential expressed proteins in total, Figure 2c).

### 2.5. Two Distinct Protein Panels Differentiate Osteoblasts and Adipocytes from Mesenchymal Stem Cells

Next, the separation of the three distinct cell types based on their protein expression profiles was analyzed. In total, 15 proteins which were both identified as differentially expressed in the OB vs. MSC and OB vs. AD comparison were defined as osteoblast-specific proteins. In analogy, 30 proteins that were differentially expressed in the AD vs. MSC and OB vs. AD comparison were defined as adipocyte-specific proteins (Figure 2d). To validate the results of the ANOVA analysis, expression data were further evaluated by computing their variable importance using machine learning algorithms (LASSO [41], elastic nets [42], random forests [43], and extreme gradient boosting [44]): the results confirmed 9 out of 15 proteins (60%) for the osteoblastic and 13 out of 30 proteins (43%) for the adipocytic panel.

#### 2.5.1. Osteoblastic Panel

A total of nine differentially expressed proteins were calculated combining two feature selection approaches (classical statistical analysis and machine learning algorithms) for the osteoblastic panel showing six (66%, AEBP1, BGN, CARMIL1, CYP24A1, MCM4, STMN1) with a higher and three (33%, COL3A1, MEST, P4HA1) with lower expression in the osteoblastic group. Expression levels of these nine osteoblast-specific proteins are visualized in Figure 3a. Closer inspection of the figure showed a protein expression of AEBP1 and MCM4 in the AD and MSC group with uniformly low expression values (range of means: 1.09–1.60) and coefficients of variation (CV, range: 0.28–0.64).

#### 2.5.2. Adipocytic Panel

For the adipocytic panel, a total of 13 differentially expressed proteins (ACSL1, CD36, EPHX1, FABP4, HP, HSD11B1, ITIH1, MAOA, PLIN1, PLIN4, PLPP1, RAP2A, SCD) were detected all being more highly expressed in the AD group. Expression levels of these 13 adipocyte-specific proteins are visualized in Figure 3b. It is apparent from this figure that the protein expression of FABP4, ITIH1, SCD, PLIN1, and PLIN4 in the OB and MSC group presented uniformly low expression values (range of means: 0.24–1.04) and coefficients of variation (CV, range: 0.36–1.72).

As a summary, differentially expressed proteins which were cell-type specifically differentially expressed and further validated by the machine learning strategies are presented as a heatmap in Figure 4 and Appendix A including gene symbols, UniProt IDs, descriptive statistics, ANOVA with post-hoc results, fold changes and functional GO-term annotations.

### 2.6. MSC Marker Protein Panel Comparison

The Mesenchymal Tissue Stem Cell Committee of the International Society of Cellular Therapy (ISCT) defined a set of cluster differentiation (CD) cell surface markers for MSC classification (CD105+, CD73+, CD90+, CD45-, CD34-, CD14-, CD11b-, CD79a- or CD19- and HLA-DR-). However, reports suggest that this marker panel is insufficient to distinguish undifferentiated MSCs from differentiated adipocytes or osteoblasts [32,38]. This inconsistency is also supported by our quantitative protein data shown in Table 2. It is apparent from this table that all ISCT-defined markers identified showed no significance across comparisons. Additionally, CD90 was the only protein to present a net log_2_ fold change >1.0 (OB vs. MSC comparison). All CD markers that are defined by a very low protein level by the ISCT were not identified in the analyzed samples.

## 3. Discussion

Therapy development for bone regeneration is highly challenging and depends on the in vitro differentiation of MSCs into favorable bone-forming osteoblasts. It has been shown that, e.g., osteoblast-like cells derived from MSCs can prevent glucocorticoid-induced bone loss [45] and support bone regeneration [46]. However, the prediction of MSCs to differentiate into osteoblasts is hampered by the incomplete biomolecular understanding and the lack of cellular biomarkers that define the quality of cells designated for therapy [29]. The present study was designed to determine differences between MSCs and their differentiated adipocytes (AD) and osteoblasts (OD) on the proteome level. Although previous studies evaluated the relationship between MSCs and osteoblasts using proteomics approaches such as mass spectrometry [47,48,49,50], this is the first report comparing the global proteome of MSCs, osteoblasts, and adipocytes by using label-free mass spectrometry in data-independent acquisition mode for quantification. Additionally, updated workflows for machine learning algorithms were applied for in-depth protein identification and data evaluation.

Cluster analysis of proteomics data indicated a clear separation of all cell types along with an enrichment of hallmark gene sets, e.g., associated with adipogenesis including, e.g., PPAR (Peroxisome proliferator-activated receptors) signaling gene sets which is considered as the key master transcription regulator in adipocytes [51]. The Mesenchymal and Tissue Stem Cell Committee of the International Society for Cellular Therapy (ISCT) has defined MSC by a set of present (+) and absent (-) cluster of differentiation (CD) markers (CD105+, CD73+, CD90+, CD45-, CD34-, CD14-, CD11b-, CD79a- or CD19- and HLA-DR-), their plastic adherence capacity and their multipotent differentiation potential when cultured in standard conditions as minimal quality [39]. However, it has been reported that positive markers of the ISCT panel are homogenously expressed among all MSC progeny [32]. Consistent with the literature, our protein analysis demonstrated that CD105, CD73, and CD90 were detectable but did not show a differential expression between MSCs, adipogenic and osteoblastic cells (Table 2). Additionally, all ISCT-defined markers that should present a very low protein abundance in MSCs have not been identified at all. These findings suggest that (a) our applied mass spectrometric workflow is capable of validating ISCT + -markers and (b) new protein markers to differentiate between MSCs, adipocytes, and osteoblasts are needed.

The model of the MSC differentiation into AD and OB allows us to determine their global proteome using quantitative mass spectrometry combined with neuronal networks and machine-learning algorithms for the first time. Interestingly, Aasebø et al. performed the only recent similar study and reported a strong separation of osteoblasts and MSCs [50] by comparing mass spectrometric data after data-dependent acquisition. However, and in contrast to our data, the proteome of adipocytes was not evaluated and only proteins were reported that showed an abundance level in the osteoblasts and not in the MSCs (*n* = 156). The findings of the here presented data considered positive and negative effect sizes (log_2_FC > |2|) resulting in 50 differentially expressed proteins between MSCs and osteoblasts. In line with the results presented by Aasebø, we detected decorin (DCN) and biglycan (BGN) as potential protein markers for osteoblastic differentiation (Appendix A): while BGN acts on the cell surface and is involved in the matrix mineralization [52,53,54], DCN was described to promote osteoblast differentiation fate [55]. It must be noted though that DCN could not be validated by machine learning algorithms and was thus not included in our final differentiation marker panel for osteoblastic differentiation.

### 3.1. Osteoblastic Panel

Overall, the applied algorithms revealed nine specific proteins for the osteoblastic differentiation, respectively. Analogous to BGN, cytochrome P450 family 24 subfamily A member 1 (CYP24A1), AE binding protein 1 (AEBP1), and collagen type III alpha 1 chain (COL3A1) concern functions of bone mineralization, osteoblastogenesis as well as matrix remodeling and thus confirmed a close association to the osteoblastic molecular differentiation module [56,57,58,59,60]. Strikingly, CYP24A1 presented the highest fold-change comparing OB to MSC (log_2_FC 3.65) and OB to AD (log_2_FC 3.52). Further, three identified proteins were reported to be associated with osteoblastic diseases: while Stathmin 1 (STMN1) was described for osteoblast and osteoclast function [61] as well as for osteopenic phenotypes in mice [62], we found minichromosome maintenance complex component 4 (MCM4) which seems to play an important role during cell division and metastasis-free survival of osteosarcoma patients [63]. Further, prolyl 4-hydroxylase subunit alpha 1 (P4HA1) has been found as the active catalytic component of the prolyl 4-hydroxylase which catalyzes the post-translational formation of 4-hydroxyproline [64,65]. High P4HA1 gene and protein expression values have been recently described as a prognostic predictor in head and neck squamous cell carcinoma [66] as well as primary melanomas [67]. Additionally, P4HA1 is associated with the collagen-dependent bone disease osteogenesis imperfecta [68,69]: as P4HA1 is involved in the post-translational modification of collagens, a direct involvement in the pathogenesis of osteogenesis imperfecta is conceivable. Noteworthy, AEBP1 and MCM4 presented low protein levels in the AD and MSC groups making AEBP1+ and MCM4+ most suited as new OB or ISCT differentiation markers with a known osteoblastic background.

No literature link for osteoblastic differentiation or diseases was found for the proteins capping protein regulator and myosin 1 linker 1 (CARMIL1), and mesoderm specific transcript (MEST). While the CARMIL1 was more highly expressed in the OB than in the MSC and AD groups, the expression pattern for MEST was reduced. The plasma-membrane-associated protein CARMIL1 plays a role in the regulation of actin polymerization and cell migration. Specifically, CARMIL1 prevents the F-actin heterodimeric capping protein (CP) activity of migrating cells and thus stimulates actin polymerization [64]. In this context, it has been shown that activation of actin polymerization decreases osteoblast differentiation and bone formation in MSCs [70]. Since we observed high CARMIL1 levels in OBs, one could assume that the endpoint of an osteoblastic differentiation process is marked by high protein levels of CARMIL1 to activate actin polymerization and thus to stop cellular differentiation mechanisms.

Last, MEST—one of the markers with a low expression in osteoblastic cells—has been described to be involved in the mesoderm development and the regulation of lipid storage. Inline, it was described as a specific protein for the endoplasmic reticulum that co-localizes within lipid droplets in cells undergoing adipogenic differentiation [71]. Additionally, elevated gene expression of MEST in preadipocytes differentiating in adipocytes was described by Kadota et al. [72].

### 3.2. Adipocytic Panel

The differentiation of MSCs into adipocytes resulted in a higher expression of 13 proteins from which 10 support the work of other studies in this area. While fatty acid-binding protein 4 (FABP4), perilipin 1 and 4 (PLIN1/4), haptoglobin (HP), and CD36 have been described to be associated with adipogenesis, hydroxysteroid 11-beta dehydrogenase 1 (HSD11B1), monoamine oxidase A (MAOA), stearoyl-CoA desaturase (SCD), and adipose acyl-CoA synthetase-1 (ACSL1) demonstrate molecular function in adipose tissues [73,74,75,76,77,78,79,80]. FABP4 presented the highest differential expression compared to OB (log_2_FC 7.04) and MSC (log_2_FC 6.77) and nearly no protein expression in the OD and MSC group. Associated with adipocytic processes, this finding could be used to implement FABP4+ as new AB or ISCT differentiation markers.

Noteworthy and to our best knowledge, no specific association to adipocyte differentiation was described for inter-alpha-trypsin inhibitor heavy chain 1 (ITIH1), RAP2A member of RAS oncogene family (RAP2A), epoxide hydrolase 1 (EPHX1), and phospholipid phosphatase 1 (PLPP1). While ITIH1 belongs to a protein family of related plasma serine protease inhibitors which is involved in extracellular matrix stabilization and the prevention of tumor metastasis [81], RAP2A is a small GTP binding protein that may regulate cytoskeletal rearrangements, cell migration, cell adhesion, and spreading [82]. EPHX1 is a member of the epoxide hydrolase family which plays a role in the metabolism of endogenous lipids such as epoxide-containing fatty acids [83] and fulfills a key function in the detoxification of xenobiotics [84]. Lastly, PLPP1 is a magnesium-independent phospholipid phosphatase of the plasma membrane and is associated with the regulation of inflammation, platelet activation, cell proliferation, and migration [85,86]. Against this background, it could be hypothesized that all four proteins may play a reasonable role in adipocytic processes. However, further studies including larger patient collectives are required.

In conclusion, this study set out to characterize undifferentiated MSCs and cells after osteoblastic and adipocytic differentiation on the proteome level. These experiments identified 22 highly cell type-specific proteins for MSCs, adipocytes, and osteoblasts using the combination of deep neural network-based quantification of data-independent mass spectrometry data. Overexpression of AEBP1 and MCM4 for OB as well as of FABP4 for AD differentiation seem to be the most promising molecular targets which could be used for regenerative medicine, stem cell, and cancer research in the future. Further studies to evaluate the molecular basis for bone formation including single-cell criteria and clinical patient data are warranted.

## 4. Materials and Methods

### 4.1. Donors and Materials

Bone marrow was aspirated from the lower extremities of five adult donors undergoing surgeries at the Department of Orthopedics and Traumatology, Odense University Hospital. The donor collective consisted of one male and four females. The bone marrow samples were considered as ‘waste material’ from routine operations and were thus collected without any extra patient risk. All donors received oral and written information and signed a consent. The project was approved by the Scientific Ethics Committee of Southern Denmark (project ID: S-20160084).

### 4.2. Cell Isolation and Culture

Bone marrow (5–10 mL) was collected into ethylenediaminetetraacetic acid (EDTA)-coated vacutainers. MSCs were isolated from the mononuclear cell population following gradient centrifugation using Lymphoprep of the bone marrow, through plastic adherence, as described previously by Stenderup et al. [87]. The cells were cultured in minimum essential medium (MEM medium) supplemented with 10% fetal bovine serum (FBS) and 1% penicillin/streptomycin (P/S) at 37 °C in a humidified incubator with 5% CO_2_. The medium was switched to MEM medium supplemented with 10% FBS, 1% P/S, 1% GlutaMAX, 1% sodium pyruvate, and 1% nonessential amino acids (S-MEM growing medium) after the first visualization of adherent cells. At 80% confluence, the cells were trypsinized and used for analysis and further cell expansion.

### 4.3. Colony-Forming Unit-Fibroblast (CFU-f) Assay

The colony-forming unit-fibroblast (CFU-f) assay was performed to assess the colony-forming capacity of cultured MSCs. The freshly isolated cells were counted in triplicates using a hemocytometer under an optical microscope and plated at a density of 1 million cells (passage 0) into three 22.1 cm^2^ Petri dishes (TPP, 93060). Standard culture conditions were used for 17 days and colonies were visualized by crystal violet staining.

### 4.4. Cell Proliferation

The cell proliferation capacity assay was performed at the first cell passage in triplicates. The cells were counted in a hemocytometer under an optical microscope, seeded (1000 cells/well) in a 6-well plate (TPP, 92006), and cultured under standard conditions. On days 1, 3, 6, 9, 12, and 15, the cells were trypsinized and counted in a hemocytometer. The proliferation capacity of the cells was measured as the area under the curve (AUC).

### 4.5. In Vitro Cell Differentiation

#### 4.5.1. Osteoblastic Differentiation

For the osteogenic differentiation, MSCs at first passage were seeded in a 4-well plate at a density of 20,000 cells/cm^2^. At 90% confluence (after 24 h), cell culture media were replaced with osteoblastic induction media containing: 10% FBS, 1% P/S, 5 mM β-glycerophosphate, 10 nM dexamethasone, 50 μg/mL vitamin C, and 10 nM vitamin D_3_. Osteoblastic induction media were replaced every 2–3 days. After 14 days, the osteoblastic differentiation was assessed by visualization of mineralized matrix formation via alizarin red staining. The cells were washed with PBS and fixed with 70% ice-cold ethanol at −20 °C for 1 h. Afterwards, the cells were washed with Milli-Q and incubated with alizarin red (pH = 4.2) for 10 min with rotation at room temperature (RT). Subsequently, the staining intensity of alizarin red was quantified using ImageJ software.

##### Alkaline Phosphatase (ALP) Activity

The alkaline phosphatase (ALP) activity is a common biochemical measure for osteoblast activity. Cells were washed with tris-buffered saline (pH 9), fixed with 3.7% formaldehyde–90% ethanol for 30 s at RT, and incubated with p-nitrophenyl phosphate (1 mg/mL) in 50 mM NaHCO_3_ and 1 mM MgCl_2_, pH 9.6 at 37 °C. After 20 min of incubation, 3 M NaOH was added to stop the reaction. Absorbance was measured at 405 nm, and ALP activity values were corrected for the number of cells in each well. The cell number was determined as a measure of cell viability and determined by incubating the cells with CellTiter-Blue for 1 h at 37 °C. The fluorescent intensity at 560/590 nm (excitation/emission) was measured in the FLUOstar Omega plate reader.

#### 4.5.2. Adipocytic Differentiation

MSCs of the first passage were plated at a density of 30,000 cells/cm^2^ in a 4-well plate for 24 h. At near full confluence, the media were replaced with adipocytic induction media containing Dulbecco’s modified Eagle’s medium (DMEM) supplemented with 10% FBS, 1% P/S, 5% horse serum, 1 μM rosiglitazone (BRL) 49,653, 3 μg/mL insulin, 100 nM dexamethasone and 225 μM 3-isobutyl-1-methylxanthine (IBMX). Media were changed every 2–3 days. After 14 days, adipocytic differentiation efficiency was determined by visualizing the formation of mature adipocytes containing lipid droplets using Oil Red O staining. The cells were fixed with 4% paraformaldehyde (PFA) for 10 min at RT, washed with 3% isopropanol, and incubated with filtered Oil Red O solution (25 mg of Oil Red O in 5 mL of 100% isopropanol and 3.35 mL Milli-Q). Photomicrographs of the differentiated cells were captured using an Olympus optical microscope (×10 magnification objective) and quantified as the area of lipid droplets (average of 6 images per sample) using ImageJ software.

### 4.6. Flow Cytometry

Flow cytometry was used for measuring the expression of surface CD markers for MSC characterization. After ex vivo expansion to passage 2, MSCs were trypsinized and washed with phosphate-buffered saline (PBS) (without Ca^2+^ and Mg^2+^) containing 2% FBS. The cells were incubated with primary fluorophore-conjugated antibodies as follows: CD14-PE, CD44-PE, CD34-PE, CD73-PE, CD90-PE, CD105-PE and ALPL-APC for 25 min at 4 °C. After the incubation, MSCs were washed twice to remove antibodies with unspecific binding and were analyzed using a BD LSR II Flow Cytometer and the BD FACSDiva software. The data were analyzed with Kaluza Flow Cytometry Analysis Software Version 1.3 (Beckman Coulter, Brea, CA, USA).

### 4.7. Cell Isolation for Mass Spectrometry Analysis

Five samples per cell lineage derived from bone marrow donors (patient numbers 11, 13, 15, 17, and 18) were selected and prepared for mass spectrometry analysis: (a) MSCs without differentiation; (b) MSCs after osteoblastic differentiation (osteoblasts); (c) MSCs after adipocytic differentiation (adipocytes). The Protease-Inhibitor (PIH) buffer was prepared as follows: 4 µL aprotinin (1:1000) + 4 µL leupeptin (1:1000) + 80 µL phenylmethylsulfonyl fluoride (PMSF) (1:50) in 3.91 mL 2-D Lysis Buffer. The cells were washed twice with 10 mL DPBS at 4 °C. Afterwards, the cells were scraped off the plate into 1.7 mL ice-cold PIH buffer centrifuged at 660× *g* for 3 min at 4 °C. The pellet was resuspended in 1 mL PIH buffer and transferred to pre-weighted cryo-tubes and centrifuged at 2700× *g* for 5 min at 4 °C. The pellet was frozen at −80 °C.

### 4.8. Two-Dimensional Fluorescence Gel Electrophoresis

The samples were assessed as described previously [88]. Briefly, samples were lyophilized and subsequently dissolved in a DIGE lysis buffer (30 mM Tris, 7M Urea, 2M Thiourea, 4% CHAPS (3-[(3-Cholamidopropyl)-dimethylammonio]-1-propansulfonat Hydrate)). Subsequently, protein samples were precipitated with the trichloroacetic acid (TCA)-like ReadyPrep 2-D Cleanup Kit (Bio-Rad Laboratories, Hercules, CA, USA) as specified by the manufacturer. Total protein concentration was determined in quadruples using the fluorescence-based EZQ Protein Quantitation Kit (Life Technologies, Carlsbad, CA, USA) according to the manufacturer’s protocol.

After labeling the protein samples with the fluorescence-based Refraction-2D Labelling Kit (NH DyeAGNOSTIC, Germany), proteins were diluted with rehydration sample buffer (7 M urea, 2 M thiourea, 2% (*w*/*v*) CHAPS, 2% (*v*/*v*) carrier ampholytes (pH 4–7) and bromophenol blue) and applied to immobilized pH gradient (IPG) gel strips, with a pH range 4–7 (Immobiline DryStrip pH 4–7, 24 cm, linear, GE Healthcare, Chicago, IL, USA). Isoelectric focusing (IEF) was carried out in a Protean i12 IEF cell (Bio-Rad Laboratories, USA) at 20 °C reaching approximately 57,700 Vh. The horizontal second dimension (HPE FlatTop Tower, SERVA Electrophoresis, Heidelberg, Germany) was carried out by sodium dodecyl sulfate-polyacrylamide gel electrophoresis (SDS-PAGE) on 12.5% acrylamide gels (2DHPE Large Gel NF 12.5% Kit, 0.65 × 200 × 255 mm, SERVA Electrophoresis, Heidelberg, Germany). Electrophoresis was performed with 1500 V for 4 h 50 min reaching approximately 3400 Vh.

Gel image acquisition was performed using the Typhoon FLA 9000 laser scanner (GE Healthcare, UK). Subsequently, protein spots were evaluated using the software Progenesis SameSpots (Nonlinear Dynamics, Newcastle upon Tyne, UK, v4.1). The analysis included protein spot detection, background subtraction, and relative quantification.

### 4.9. Sample Preparation for High-Performance Liquid Chromatography (HPLC) and Electrospray Ionization Tandem Mass Spectrometry (ESI-MS/MS)

The sample preparation for mass spectrometry analysis was performed with a filter aided sample preparation (FASP) protocol [89]. Briefly, 100 µg samples diluted in 200 µL 8M uric acid in 0.1 M Tris-HCL (UA) were added to filter columns (30 k, AmiconUltra, Merck, Darmstadt, Germany) and centrifuged at RT, 14,000× *g* for 15 min. Additional 200 µL UA was added to the filter column and repeatedly centrifuged. After discarding the eluate, 100 µL 0.05 M IAA (solubilized in UA) was added to the filter column and incubated in the dark at RT for 20 min. The column was washed two times with UA before 100 µL 0.05 M ammonium bicarbonate (ABC) buffer was added to the columns and centrifugation at RT, 14,000× *g* for 10 min, twice. Next, 40 µL trypsin in ABC (enzyme:protein ratio 1:100) was added and left for incubation in a humid chamber at 37 °C overnight. The columns were centrifuged at RT (14,000× *g* for 10 min) before 40 µL of ABC were added. Finally, the filtrate was collected, lyophilized, and stored at −20 °C after centrifugation at 14,000× *g* for 10 min.

### 4.10. High-Performance Liquid Chromatography (HPLC) and Electrospray Ionization Tandem Mass Spectrometry (ESI-MS/MS)

The samples were solubilized with a final concentration of 1 µg/µL in solvent A (0.1% formic acid) and were loaded into a HPLC Dionex Ultimate 3000 (Thermo Fisher Scientific, Waltham, MA, USA). The samples were first loaded onto a trap column (μ-Precolumn Acclaim PepMap100, internal diameter: 0.3 × 5 mm, 5 μm, 100 Å, Thermo Fisher Scientific, Waltham, MA, USA) and desalted with loading solution at 10 μL/min for 4 min. Peptides were subsequently separated using an analytical column (LC Column, 3 μm C18 (2), 0.3 × 50 mm, 3 μm, 100 Å, Phenomenex Inc., Torrence, CA, USA) and eluted with a multi-step gradient of solvent B (0.1% formic acid in acetonitrile) in solvent A for 86 min at a flow rate of 5 µL/min. Purified peptides were analyzed with a TripleTOF 5600+ mass spectrometer (AB ScieX, Framingham, MA, USA). The following SWATH (sequential window acquisition of all theoretical mass spectra) acquisition working parameters were used: Ion Spray Voltage Floating (ISVF) at 5000 V; ion source gas (GS1), 15; ion source gas (GS2), 0; curtain gas (CUR) at 30 and source temperature heating set to 0 °C. The optimized declustering potential (DP) was set at 100; collision energy (CE) to 19.2; collision energy spread (CES), 5.0; ion release delay (IRD), 67; ion release width (IRW) at 25. For data acquisition, one 0.049965 s MS scan (m/z 350–1250) was performed, followed by 100 variable Q1 windows with the size range 5–91.3 Da, each at 0.030 s accumulation time with CES at 5 eV. The precursor isolation windows were defined using the SWATH Variable Window Calculator V1.1 (AB Sciex) based on precursor *m*/*z* densities obtained from DDA spectra. For DDA acquisition, identical instrument working parameters were used. MS scans were performed for 350–1250 Da with an accumulation time of 0.25 s, MS/MS scans were performed for 100–1500 Da with an accumulation time of 0.05 s at high sensitivity mode.

### 4.11. SWATH Data Processing

The raw SWATH data were processed using the software tool DIA-NN v1.7.16 (data-independent acquisition by neural networks) developed by Vadim Demichev et al. [90]. The software was used in the high accuracy LC mode with RT-dependent cross-normalization enabled. Mass accuracy, MS1 accuracy, and scan window settings were set to 0, as DIA-NN optimizes these parameters automatically. The ‘match between runs’ function was used to first develop a spectral library using the ‘smart profiling strategy’ from the data-independent acquisition data. The human UniProtKB/swiss-prot database (version 2020/12/6) [91] was used for protein inference from identified peptides. Trypsin/P was specified as protease. The precursor ion generation settings were set to peptide length of 7–52 amino acids, the maximum number of missed cleavages to one. The maximum number of variable modifications was set to zero. N-terminal methionine excision and cysteine carbamidomethylation were enabled as fixed modifications. The neural network classifier was set to double-pass mode as it typically generates the best results and analysis time was not an issue here. The resulting report file was further processed in the DIA-NN R package [90] for MaxLFQ-based [92] protein quantification. A report was generated containing unique proteins (proteins that were not assigned to a group of homologs) that passed the FDR cut-off of 0.01 applied on the precursor level and were identified and quantified using proteotypic peptides only. All proteins in the final dataset were identified by at least two unique peptides. The proteins were mapped for their corresponding gene names, which were required for downstream analysis steps such as gene enrichment analysis. In this context, the terms proteins and genes are used interchangeably in this study report.

The mass spectrometry proteomics data have been deposited to the ProteomeXchange Consortium [93] via the PRIDE [94] partner repository with the dataset identifier PXD029900.

### 4.12. Quantitative Data Processing

The dataset was further processed with the R [95] package DEP [96], which allows for missing value filtering and imputation: proteins/genes that were not reliably identified in 4 out of 5 replicates of at least one condition were removed from the dataset. Variance stabilizing normalization was applied to the remaining dataset using the R package vsn [97]. To analyze all identified proteins, remaining missing values were imputed with a multiple imputation strategy, as those can conserve differential expression, maintain the original informational content of the dataset, and respect low concentrated/not detectable proteins [96,98]. Missing values were considered as missing not at random (MNAR) when protein quantity data were completely missing for at least one condition. All other missing values were considered missing at random (MAR). While MNAR were imputed with random draws from a Gaussian distribution centered around a minimal value, MAR were imputed with a k-nearest neighbor model. The imputed dataset was used for subsequent statistical analysis.

### 4.13. Data Analysis

The gene name of the corresponding protein was used for analyses. The bioinformatics platform Omics Playground v2.7.18 (BigOmics Analytics, Lugano, Switzerland) was partly used for protein quantity data analysis and visualization [99]. For a clustering analysis and visualization of the high-dimensional quantitative proteomics data, a principal components analysis plot was computed using the stats package provided by the R software suite [95], and visualized with the R package ggplot2 [100]. A heatmap (two-way hierarchical clustering) was computed in Omics Playground. Using the R package NbClust [101], we utilized several algorithms for the estimation of the adequate cluster numbers within the data set of the top 50 differentially expressed proteins (selected via their standard deviation). According to a majority vote of the NbClust algorithms, three is the best number of clusters for this data set. Thus, ‘three clusters’ was used as a parameter for the initialization of the unsupervised two-way hierarchical clustering method included in the Omics Playground software suite. The method was applied to the data set of the top 50 proteins and used to generate the heatmap.

The gene set enrichment was computed with Omics Playground using the merged results of the Fisher’s exact test [102], fGSEA [103], and GSVA [104], thereby applying a FDR < 0.05 (which corresponds to a so-called ‘meta.q value’ which corresponds to the highest q-value provided by the used statistical methods) and absolute |log_2_FC| threshold of >0.2. The MSigDB gene set collection was used as the target database for the enrichment. All measured genes are used as ‘universe’ after filtering for non-expressed genes. The log_2_FC is calculated as the average log_2_FC of all genes identified in the particular gene set.

For a differential protein expression analysis, a Welch-ANOVA was computed using the stats package. The results of the Welch-ANOVA were corrected for multiple testing via the Benjamini–Hochberg procedure [105], selecting the candidates for post-hoc testing with a FDR of <0.05. Post-hoc testing was performed via the Tukey honestly significant difference (HSD) method using the TukeyHSD function of the R software suite. Significant differential protein expression was considered at a *q*-value of <0.05 and an absolute logarithmic fold change (|log_2_FC|) of >2. Volcano plots were created from the results of the differential expression analysis, using the plot function in R. For better visualization, the -log_10_ q-values of the ANOVA were plotted on the *y*-axis and log_2_FC on the *x*-axis. A Venn diagram was created using the web tool InteractiVenn [106], using the differentially expressed proteins of the three comparisons as input sets. Boxplots for overlapping proteins were created in the R software suite using ggplot2. The ‘biomarker’ module of Omics Playground was used for the ranking of biomarkers that could be suitable for the characterization of the cell lineage. Here, a cumulative variable importance score for each feature was calculated using machine learning algorithms, including LASSO [41], elastic nets [42], random forests [43], and extreme gradient boosting [44] and provides the top 40 features according to the ranking of the cumulative score by the algorithms. The findings of the machine learning algorithms were used as further validation of our marker panels derived from classical statistical testing. The Panther classification system was used for the annotation of all overlapping proteins according to gene ontology terms of the biological process, molecular function, and cellular component [107].

### 4.14. Reagents

Lymphoprep (StemCell Technologies, Vancouver, Canada, 1114545), minimum essential media (MEM, Gibco/Thermo Fisher Scientific, Waltham, MA, USA, 31095-029), Dulbecco’s modified Eagle’s medium (DMEM, Gibco/Thermo Fisher Scientific, 31966), fetal bovine serum (Thermo Fisher Scientific, Waltham, MA, USA, 10270106), GlutaMAX (Gibco/Thermo Fisher Scientific, 35050-038), non-essential amino acids (MEM NEAA, Gibco/Thermo Fisher Scientific, 11140-035), Trypsin-EDTA (Invitrogen/Thermo Fisher Scientific, 25300062), β-glycerophosphate (Calbiochem/Merck, Darmstadt, Germany, 35675), dexamethasone (Sigma/Merck, Darmstadt, Germany, D4902), vitamin C (L-Ascorbic Acid Phosphate Magnesium Salt n-Hydrate, Wako, Neuss, Germany, 013-12061), vitamin D3 (1α,25-Dihydroxyvitamin D3 a kind gift from Leo Pharma, Ballerup Sogn, Denmark), p-nitrophenyl phosphate (Sigma/Merck, 71768), Alizarin Red (Sigma/Merck, A5533), Oil Red O (Sigma/Merck, O0625), horse serum (Sigma/Merck, H1270), rosiglitazone (BRL, Cayman Chemical, Ann Arbor, MI, USA, 71740), insulin (Sigma/Merck, I9278), 3-isobutyl-1-methylxanthine (IBMX, Sigma/Merck, I5879), Napthol AS-TR phosphate disodium salt (Sigma/Merck, N6125), Fast Red TR Salt hemi (zinc chloride) salt (Sigma/Merck, F8764), anti-CD14 (BD Pharmingen, Franklin Lakes, NJ, USA, 555398), anti-CD44 (Beckman Coulter, Brea, CA, USA A32537), anti-CD34 (BD Biosciences, Franklin Lakes, NJ, USA, 555822), anti-CD73 (BD Bioscience, 550257), anti-CD90 (Beckman Coulter, IM3600U), anti-CD105 (Beckman Coulter, A07414), anti-ALPL (R&D Systems, Minneapolis, MN, USA, FAB1448A), CellTiter-Blue cells viability assay reagent (Promega, Walldorf, Germany, G8081), TRIzol (Invitrogen/Thermo Fisher Scientific, 15596018), High-Capacity cDNA Reverse Transcription Kit (Applied Biosystems, 4368813), Fast SYBR Green Master Mix (Applied Biosystems/Thermo Fisher Scientific, Waltham, MA, USA 4385614), RNAse (Affymetrix, Santa Clara, CA, USA, 78020Y), DNAse (Worthington, Columbus, OH, USA, DPRF), Tris (Merck, 1083821000), uric acid (Affymetrix, 75826), thiourea (Sigma/Merck, 33717), CHAPS (Sigma/Merck, C5070-1G), ReadyPrep 2-D Cleanup Kit (Bio-Rad Laboratories, Hercules, CA, USA, 163-21-30), Tris-HCL (Roth, Karlruhe, Germany, 48551), ammoniumbicarbonate (Honeywell, Morristown, NJ, USA, 1066-33-7), iodacetamide (Sigma/Merck, 163-2109), Trypsin gold (Promega, V5280), formic acid (Baker, 9820), acetonitrile (Baker/Thermo Fisher Scientific, 9017).

## Figures and Tables

**Figure 1 ijms-23-02568-f001:**
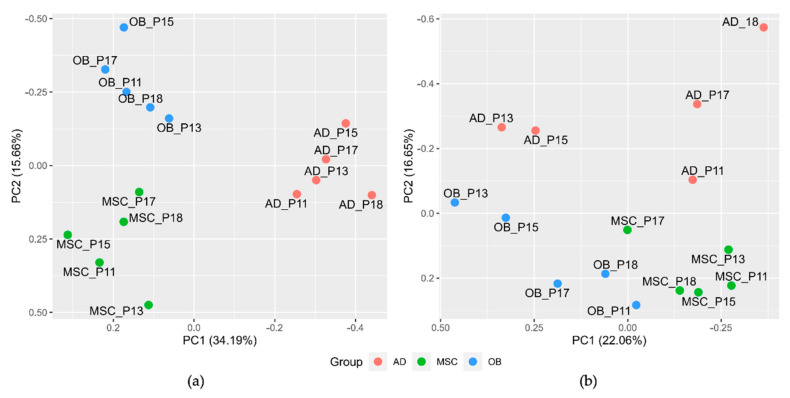
Unsupervised principal component analysis plots derived from mass spectrometry data ((**a**), 4108 proteins) and two-dimensional gel electrophoresis data ((**b**), 1517 protein spots). The three cell lineages MSCs (green), OB (blue), AD (pink) were run in quintuples. P11, p13, p15, p17, and p18 indicate the patient number of the corresponding sample. *X*- and *y*-axes show the first and second principal components, respectively.

**Figure 2 ijms-23-02568-f002:**
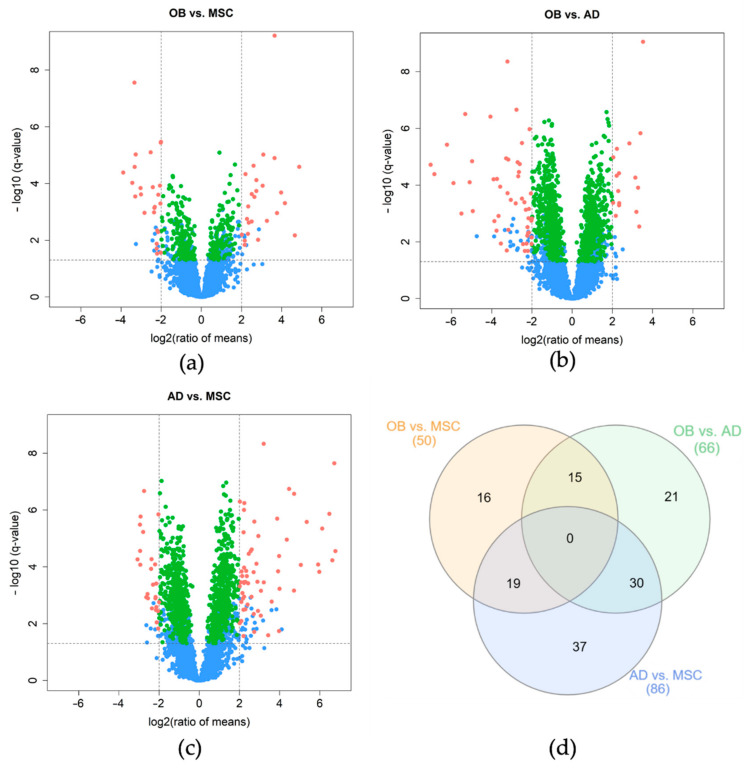
Volcano plots of differentially expressed proteins between MSCs, osteoblasts, and adipocytes (**a**–**c**) as well as a Venn diagram indicating the overlap of differentially abundant proteins between all three group comparisons (**d**). Volcano plots are presented with the fold-change of the corresponding comparison in logarithmic scale (*x*-axis) against the q-value of the Tukey’s post hoc test (*y*-axis). Significance thresholds (q-value < 0.05 and |log_2_FC| threshold of >2) are indicated by dashed lines. Proteins passing these cut-offs are considered significant and colored in pink. Proteins passing the ANOVA and post-hoc q-value but not the log_2_FC threshold are colored in green. Proteins that were not significant in the ANOVA but in Tukey’s post hoc are indicated in blue. (**a**) OB vs. MSC comparison, 50 proteins are identified as significantly differentially expressed. (**b**) OB vs. AD comparison, 66 proteins are identified as significantly differentially expressed. (**c**) AD vs. MSC comparison, 86 proteins are identified as significantly differentially expressed. (**d**) Venn diagram indicating the overlap of differentially abundant proteins (ANOVA) of all three defined comparisons.

**Figure 3 ijms-23-02568-f003:**
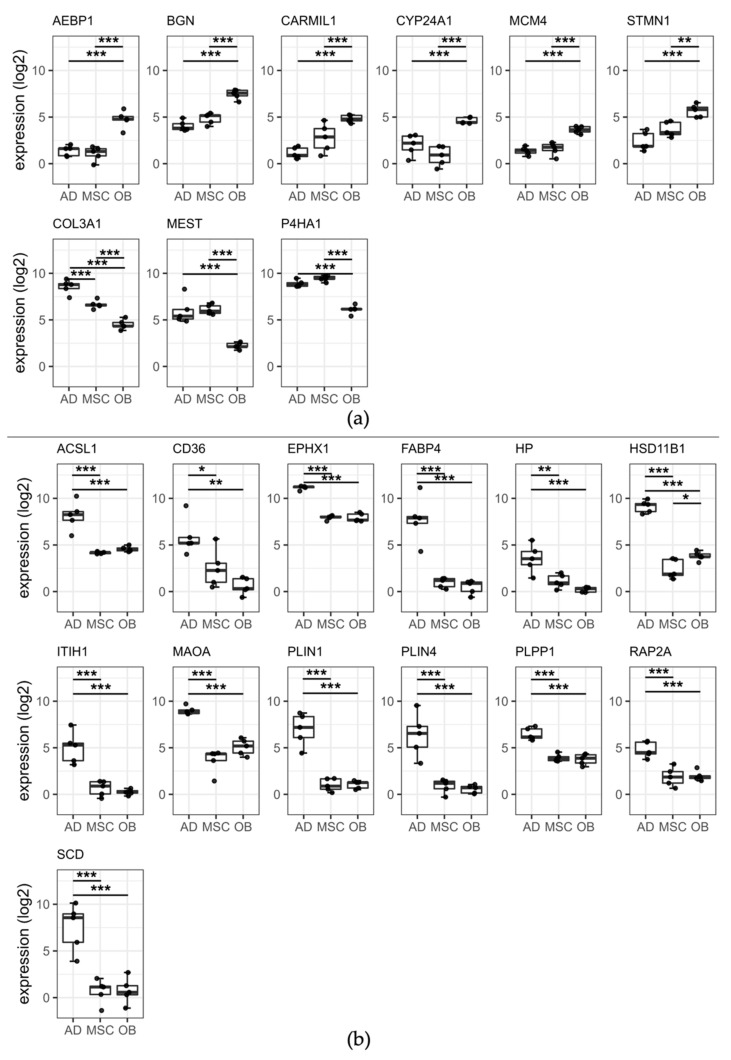
Boxplots of nine proteins for (**a**) osteoblastic differentiation and (**b**) adipocytic differentiation using ANOVA and machine learning algorithms. Individual data points are shown with median expression in log_2_ of the measured intensity. Whiskers are plotted according to the Tukey method indicating 1.5 * interquartile range. The osteoblastic protein (**a**) COL3A1 and the adipocytic protein (**b**) HSD11B1 show significant differential expressions for all three comparisons (indicated by asterisks), however, the comparison between the AD and MSC group and OB and MSC did not exceed the net log_2_FC threshold of 2, respectively. Asterisks indicate the Tukey’s post hoc q-value results: * =≤ 0.05, ** =≤ 0.01, *** =≤ 0.001.

**Figure 4 ijms-23-02568-f004:**
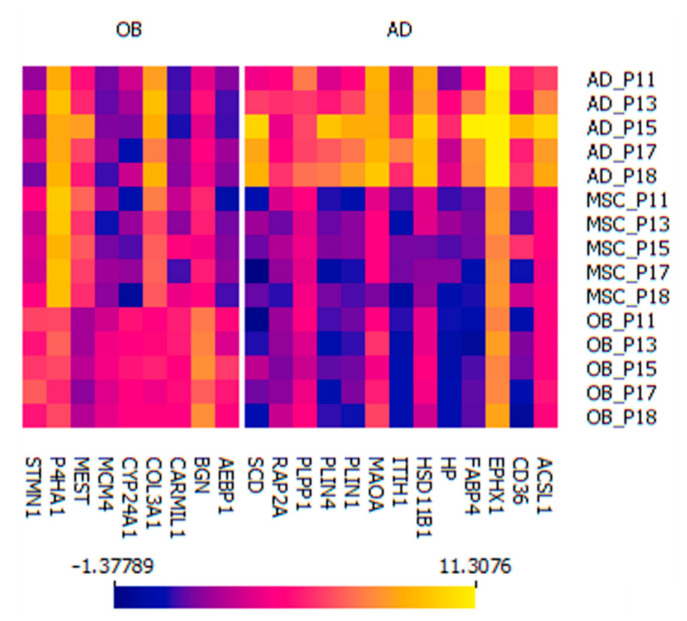
Heatmap of 22 marker proteins for osteoblastic and adipocytic differentiation. Expression values are shown in log_2_ of the measured intensity.

**Table 1 ijms-23-02568-t001:** Enriched gene sets of the MSigDB Hallmark collection for OB versus AD, OB versus MSC, and AD versus MSC. (Log_2_FC, logarithm of fold change; Meta-q, meta q-value used statistical methods; Avg. Expr., average expression).

Gene set	Log_2_FC OB vs. AD	Meta-q	Avg. Expr. OB	Avg. Expr. AD
MYC_TARGETS_V1	0.575	0.0005	6.103	5.528
E2F_TARGETS	0.410	0.0005	7.049	6.638
G2M_CHECKPOINT	0.278	0.0005	5.411	5.133
PEROXISOME	−0.223	0.0366	6.925	7.148
BILE_ACID_METABOLISM	−0.317	0.0006	6.762	7.079
FATTY_ACID_METABOLISM	−0.350	0.0006	6.535	6.885
CHOLESTEROL_HOMEOSTASIS	−0.377	0.0315	6.318	6.695
OXIDATIVE_PHOSPHORYLATION	−0.438	0.0006	5.028	5.466
ADIPOGENESIS	−0.491	0.0006	5.782	6.273
**Gene set**	**Log_2_FC OB vs. MSC**	**Meta-q**	**Avg. Expr. OB**	**Avg. Expr. MSC**
MYC_TARGETS_V2	0.279	0.0435	5.745	5.466
E2F_TARGETS	0.274	0.0008	7.034	6.76
MYC_TARGETS_V1	0.241	0.0098	6.076	5.835
**Gene set**	**Log_2_FC AD vs. MSC**	**Meta-q**	**Avg. Expr. AD**	**Avg. Expr. MSC**
OXIDATIVE_PHOSPHORYLATION	0.659	0.0006	5.590	4.931
CHOLESTEROL_HOMEOSTASIS	0.533	0.0006	6.654	6.121
ADIPOGENESIS	0.529	0.0006	6.206	5.677
FATTY_ACID_METABOLISM	0.478	0.0006	6.909	6.431
BILE_ACID_METABOLISM	0.411	0.0006	7.096	6.685
PEROXISOME	0.337	0.0028	7.282	6.945
XENOBIOTIC_METABOLISM	0.271	0.0011	5.857	5.585
PI3K_AKT_MTOR_SIGNALING	−0.203	0.0359	7.127	7.331
MYC_TARGETS_V1	−0.334	0.0105	5.605	5.939
MITOTIC_SPINDLE	−0.335	0.0006	6.635	6.97

**Table 2 ijms-23-02568-t002:** Protein expression values of CD markers that have been defined by the International Society of Cellular Therapy for MSC classification. (+) defined high protein expression in MSCs; (-) defined low expression in MSCs. (CV, coefficients of variation; q-value, adjusted post-hoc *p*-value; Log_2_FC, logarithm of fold change).

Protein	Gene Name	CV	q-Value	Log_2_FC AD vs. MSC	Log_2_FC OB vs. MSC	Log_2_FC OB vs. AD
CD105+	ENG	0.104	0.25	0.121	−0.884	−0.763
CD73+	NT5E	0.064	0.43	0.426	0.157	0.583
CD90+	THY1	0.121	0.41	0.355	−1.046	−0.691
CD45-	PTPRC	Not Identified				
CD34-	CD34	Not Identified				
CD14-	CD14	Not Identified				
CD11b-	ITGAM	Not Identified				
CD79a-	CD79a	Not Identified				
CD19-	CD19	Not Identified				
HLA-DR-	HLA-DR	Not Identified				

## Data Availability

The mass spectrometry proteomics data have been deposited to the ProteomeXchange Consortium [93] via the PRIDE [94] partner repository with the dataset identifier PXD029900.

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
