# Peer review of "Protein Expression of AEBP1, MCM4, and FABP4 Differentiate Osteogenic, Adipogenic, and Mesenchymal Stromal Stem Cells"

_ijms, 2022, doi:10.3390/ijms23052568_

Round 1

Reviewer 1 Report

The work is really good. I accept the manuscript

Author Response

We thank the reviewer for considering our manuscript suitable for publication.

Reviewer 2 Report

Please find hereby my review of the article untitled “Protein expression of AEBP1, MCM4 and FABP4 differentiate osteogenic, adipogenic and mesenchymal stromal stem cells.” by Thorben Sauer et al. in Int. J. Mol. Sci.

The submitted manuscript has a very interesting topic and a real scientific interest.

The aim of the study was to characterize differences between MSCs and cells after adipogenic (AD) or osteoblastic (OB) differentiation at the proteome level. Comparative proteomic profiling was performed using tandem mass spectrometry in data-independent acquisition mode. Protein expression profiling identified activation of the Peroxisome proliferator-activated receptors (PPARs) signaling pathway after AD. In addition, two distinct protein marker panels could be defined for osteoblastic and adipocytic cell lineages. Hereby, overexpression of AEBP1 and MCM4 for OB as well as of FABP4 for AD were detected as the most promising molecular markers

The manuscript needs to be edited carefully (typographical errors…).

Why check differentiation in OB and AD but not Chondrocyte?

Discuss the interest to check veracity of the novel biomarkers in other populations of MSC from other tissues

Very beautiful paper with big techniques and strong bioinformatic methods

Author Response

The manuscript needs to be edited carefully (typographical errors…).

We agree with the reviewer and revised the manuscript carefully.

Why check differentiation in OB and AD but not Chondrocyte?

We thank the reviewer for this interesting comment and agree that differentiation into chondrocytes would have strengthened the current study. However and due to the limited number of cells obtained, it was not possible to perform chondrogenic differentiation. Additionally, the experimental design only focused on the differentiation into osteoblasts and adipocytes to further support our research agenda on studying molecular mechanisms of bone formation.

Against this background, in vitro chondrocyte differentiation was not in the scope of the study but might be considered in further validation steps.

Discuss the interest to check veracity of the novel biomarkers in other populations of MSC from other tissues.

We thank the reviewer for this interesting point. The study aimed to characterize the complex transformation process of mesenchymal stem cells into adipocytes (AD) and osteoblasts (OB) at the proteome level. Comparative proteomic profiling was used to identify cell-type-specific proteins as well as potential new differentiation markers. Hereby, we identified FABP4, AEBP1, and MCM4 as highly cell-type-specific proteins characterizing differentiated ADs and OBs, respectively.

We do agree with the reviewer that further validation studies need to be performed e.g. in other populations of MSCs for a successful clinical implementation. However, this future study goes beyond the scope of the work presented here and we stated that further studies are warranted on page 12. Even so, we are strongly convinced that the here reported proteins are important targets since an additional evaluation using the Human Protein Atlas revealed a remarkable tissue specificity of MCM4 and FABP4. We added this information on pages 11 and 12.